# Evaluating the Counseling Standards and Ability of Pharmacy Staff to Detect Antibiotic-Drugs Interactions: A Simulated Client Study from Pakistan

**DOI:** 10.3390/antibiotics12050931

**Published:** 2023-05-19

**Authors:** Muhammad Majid Aziz, Muhammad Fawad Rasool, Muteb Alanazi, Tareq Nafea Alharby, Jowaher Alanazi, Bader Huwaimel

**Affiliations:** 1Department of Pharmacy Practice, Faculty of Pharmacy, Bahauddin Zakariya University Multan, Multan 60000, Pakistan; pharmajid82@yahoo.com (M.M.A.); fawadrasool@bzu.edu.pk (M.F.R.); 2Department of Clinical Pharmacy, College of Pharmacy, University of Ha’il, Hail 81442, Saudi Arabia; 3Department of Pharmacology and Toxicology, College of Pharmacy, University of Ha’il, Hail 81442, Saudi Arabia; js.alanzi@uoh.edu.sa; 4Department of Pharmaceutical Chemistry, College of Pharmacy, University of Ha’il, Hail 81442, Saudi Arabia; b.huwaimel@uoh.edu.sa; 5Medical and Diagnostic Research Center, University of Ha’il, Hail 55473, Saudi Arabia

**Keywords:** simulated client, community pharmacies, patient’s counseling, communication skill, antibiotics, Pakistan

## Abstract

Effective and safe medication use can be maximized by providing medication counseling, which aims to optimize therapeutic results. This approach improves the effectiveness of antibacterial treatment, reduces treatment expenses, and mitigates the emergence of antimicrobial resistance. No research from Pakistan has been previously documented. The purpose of this research was to evaluate both the quality of antibiotic counseling provided and the level of understanding exhibited by pharmacy employees with regard to interactions involving antibiotic medications. Using a simulated client method, two scenarios were used to assess 562 pharmacies that were systematically selected. Scenario 1 focused the counseling for use of prescribed medicines with non-prescribed antibiotics. Scenario2 indicated counseling provision for prescribed antibiotics that have possible antibiotic–drug interactions. The evaluation of counseling skills was also conducted. The analysis involved the use of descriptive statistics and chi-square tests. Only 34.1% of simulated clients received medication counseling directly; 45% received it on request. About 31.2% of clients were referred to a physician without counseling. The most frequently provided information was therapy dose (81.6%) and duration (57.4%). More than half (54.0%) of clients were asked about disease duration, but drug storage was ignored. Insufficient information was provided about side effects (1.1%) and antibiotic–drug interactions (1.4%). Most (54.3%) clients were instructed about dietary or lifestyle modifications. Only 1.9% of clients received information about drug administration route. No information was provided about other medication during therapy, effect of medicine withdrawal, and compliance to medication. The current level of antibiotic counseling within Pakistani community pharmacies is inadequate and requires the attention of medical authorities. Professional training of staff could improve counseling.

## 1. Introduction

Medication counseling (MC) is a crucial aspect of ensuring high-quality of pharmaceutical care. It guarantees that patients obtain secure and efficient medicine to enhance their therapeutic outcome [1]. MC can decrease the risk of adverse events, enhance adherence to treatment, and lessen the financial burden of illnesses and complications [2]. Inappropriate medication without counseling can heighten the risk of co-morbidities and mortality [3,4]. MC has become a priority service in modern community pharmacies [2,5]. It ensures that individuals receive effective and safe medication to maximize therapeutic outcome [1]. Inappropriate medication use without counseling increases the risk of co-morbidities and mortality [6]. Many guidelines have been established that provide adequate recommendations for MC for both over the counter and prescribed only medicines [7].

MC encompasses the provision of oral or written information to patients or their guardians about the safe use of medicines, appropriate storage, possible side effects, healthy food and lifestyle modifications [8]. MC services are typically offered when medicine is dispensed or during prescription handling, but can also occur separately in modern pharmacies [6,9]. MC has become a mandatory professional responsibility of pharmacists in many developed countries. However, it is not well established in the developing world, as community pharmacies in developing countries focus only on drug trading [5,7,10,11].

An appropriate counseling greatly influences the use of antibiotics [4]. The counseling is useful tools to optimize antibacterial therapy, decrease the cost of treatment and prevents from rise ofantimicrobial resistance (AMR) [12]. In 2014, World Health Organization (WHO) stated that community pharmacies are ideallypositioned to promote the proper use of antibiotics as these are most often and first point of patients contact. Through good pharmacy practice (GPP), these can play an essential role to combat AMR [13]. These also certainly play a central role to encourage the patient for adherence to antibiotics regimens [14]. On other hand, antibiotics-drugs interactions (ADIs) are usually associated with antibiotic therapies in co-morbidities and poly-pharmacy [15,16]. Therefore, antibiotic use in pediatrics and geriatric need special consideration [17,18]. Pharmacists also have the ability to minimize ADIs [19].

The healthcare system in Pakistan is largely comprised of the private sector, which makes up a significant 79% of the overall system. Additionally, a vast majority of healthcare resources—approximately 77%—are dedicated towards medications. The poly-pharmacy is a common phenomenon [20,21]. About 80% of medicines are sold through 63,000 community pharmacies [22]. Sale of antibiotics from pharmacies is alarming [20]. Two significant obstacles to achieving GPP are the lack of pharmacists in pharmacies and inadequate education in patient care [22,23]. As a result, sub-therapeutic doses of antibiotics were given out by the community pharmacies [22,24,25]. Currently, no research has been conducted examining the quality of MC regarding antibacterial treatments in community pharmacies in Pakistan. Additionally, there is a lack of information in regards to the understanding of ADIs among community pharmacy employees. Therefore, we evaluated how pharmacies counsel the use of antibiotic to patient with prescription of ADIs and patients without prescription. This initial analysis will serve as a basis for guidance in enhancing and reinforcing antibiotic counseling.

### Theories and Conceptual Framework 

There is currently no established or agreed-upon explanation or description of counseling in literature. There are different interpretations of this concept among researchers. Some of them define it on operational basis and some on outcome [10]. In an earlier publication from the previous decade, Shah and co-author [26] referred to it as “giving of advice and provision of information”. Later, Puspitasari et al. [27] described that counseling is a beneficial practice that enhances the standard of patient care by recognizing and addressing issues related to medicine. It enables the patient to adopt self-management behaviors that are beneficial.

The counseling of the patients had been subjected to numerous changes. It was initially originated from Shannon and Weaver’s “Transmission model” of communication. Later, it was conceptualized as a two-way communication process “Transaction model”. But currently used “Ritual model” is considered most effective model of counseling [26] (Figure 1).

Various methods are used to analyze counseling services in the community pharmacies. These methods include telephonic interviews, mail surveys, observational studies, simulated patient or client method and audio analysis [28]. Simulated client (SC) method has been used in LMICs to a greater extent than developed world due to versatility in practices [29].

## 2. Methods

SC method was used in selected community pharmacies in Punjab, Pakistan. This method has been widely used to assess MC in community pharmacies [2,3,25,26]. This cross-sectional study was performed between June 2016 and February 2017.

### 2.1. Simulated Clients

Twenty-one SCs were selected. All SCs were male, aged 24–37 years, and were either final year Pharm D students or had bachelor’s level qualifications. Each SC conducted only one scenario in different pharmacies. Every SC had a good command of the local languages in the urban or rural area in which they conducted the scenarios and dressed appropriately for the areas. All SCs were trained well.

### 2.2. Communication Skills Assessment

Counseling communication skills were assessed by SCs, who rated them on a 5-point Likert scale: very poor (a score of 1), poor (2), moderate (3), good (4), and very good (5).

### 2.3. Pilot Study

Before collecting the study data, a pilot study was conducted in 21 pharmacies. The reliability of the methodology and feedback record form was assessed. To ensure equal data quality across SCs, rater reliability was analyzed using the percentage agreement method. Three experts including a senior community pharmacist, a pharmacy academician and sociologist assessed the scenario presentation of SC. Their performance was rated and scored. The feedback record form was modified based on the pilot study results. The pilot study data was not included in the final results. 

### 2.4. Documentation of Feedback

Every SC was advised to complete a feedback record form immediately after their visit to each pharmacy. The feedback record (S1) form was a partially modified version of that used in previous studies [2,3,19,20]. The content of the counseling (e.g., advice given, questioning, and counseling duration) was recorded first. Then other relevant information, such as communication skills and pharmacy-related characteristics, was recorded.

### 2.5. Study Setting

Study sites were community pharmacies in Punjab, Pakistan. The area of the Punjab province is 205,344 square kilometers and is the most populous province. The population is estimated to be more than 91 million, or 56% of the total national population [20].

### 2.6. Pharmacy Selection

A stratified sampling technique was used to select pharmacies. Nine strata were formed based the Punjab governmental administrative divisions. Each stratum was further divided into four substrata: divisional city, district city, tehsil city, and suburban and rural area.

A list of pharmacies was obtained from Department of Health [30]. After confirming each pharmacy’s licensure, pharmacies were arranged geographically with a serial number. Pharmacies were systematically selected from the list by this serial number. The availability of medicine during the pilot study was also considered when determining the sample size. The sample size of pharmacies was calculated for response distribution (60%), confidence interval (95%) and margin of error (4%) for the total 22,319 pharmacies by Raosoft [31]. The response distribution 60% was applied due to probability of distribution variable and possible variability in responses. To determine the maximum accuracy in sample margin of error was 4%. A final total of 562 pharmacies were selected (S2). We tried to ensure a homogeneous and uniform presentation of pharmacies from all areas of Punjab province.

### 2.7. Scenarios

To assess MC in community pharmacies, two different scenarios were designed, as given below:

#### Scenario 1

SC visited the pharmacy and requested medicines by mentioning the brand names, as follows: 

“I need Medicine A (Ciprofloxacin 500 mg) tablets and one pack of Medicine B (calcium + vitamin C) effervescent tablets”.

SC provided the following information to pharmacy staff on inquiry: “I need these medicines for my 73-year-old grandmother. She was diagnosed with osteoporosis a year ago and prescribed a tablet of Medicine B Plus daily with Medicine C (Nimesulide 100 mg). Now she has a sore throat. I want to purchase Medicine A for her, on the basis of my previous experience.”

If the pharmacy staff did not ask for the above information, the SC actively disclosed it when leaving and asked the following

“Can she take Medicine A at any time? Should she take Medicine A before/after meals? Can she take Medicine A, Medicine C, and Medicine B at same time?”

The significance of scenario 1: This scenario illustrates the use of prescribed medicines with non-prescribed antibiotics and the possible resultant drug–drug interactions. It also focuses on counseling for a patient’s guardian and on geriatric care with poly pharmacy in community pharmacies.

#### Scenario 2

SC visited a pharmacy with a complete prescription for a child of 9 years diagnosed with asthma. The prescription was “Theophyline 200 mg + Salbutamol respirator + and Clarithromycin suspension 125 mg/30 mL”.

If the medications were dispensed without counseling, SC requested information about the use of the medicines when leaving the pharmacy.

The significance of scenario 2: This scenario indicated poly-pharmacy counseling provision for prescribed medicines along with a potent antibiotic that have possible drug–drug interactions. This scenario illustrated the quality of counseling provided for treatment of a chronic disease. It also focused on pediatric care and the drug administration route. 

### 2.8. Ethical Considerations 

The study design and protocols were approved by the Pharmacy Research Ethics Committee at The Islamia University of Bahawalpur, Pakistan (Ref # 67-2015/PREC). The confidentiality of outcomes was also maintained by the oath of all data collectors in view of research ethics. None of the data contained identifying information; pharmacies were assigned identification numbers that were used in the data analysis process.

### 2.9. Data Analysis

The Statistical Package for the Social Sciences (SPSS) version 18.0 was used for descriptive analysis of the data related to counseling. Chi-square tests were used to determine the influence of different factors on counseling. 

## 3. Results

A total of 554 pharmacies (98.5%) responded to the SCs in scenario 1 and 465 pharmacies (82.7%) responded to scenario 3. These figures reflected the availability of medicines. In some pharmacies, the requested medicines were not available: scenario 1: 8 (1.4%) and scenario 2: 97 (17.2%). Most (60.0%) visited pharmacies had a single counter. All retailers were male and it was estimated that many (55.0%) were aged 25–45 years. Most pharmacies (97.9%) failed to provide privacy during counseling (Table 1).

Only 3.6% of pharmacies provided counseling to clients without being asked (Table 2).

Staff of less than half the pharmacies (45.9%) asked about the duration of disease or therapy. Few pharmacies (2.5%) ensured that the client received the desired information (Table 3).

Most pharmacies (81.6%) provided information related to dose of therapy, but no clients were counseled about the storage of medicine (Table 4).

The communication skills of the person providing counseling were determined by including only cases in which the staff both asked and answered questions. Attention to the customer was the skill ranked most highly (2.4 ± 0.4) and the provision of written information was the skill ranked lowest (1.1 ± 0.3) by the SCs (Table 5).

However, responses were different in each scenarios, the analysis indicated that the retailer’s age and number of waiting customers influenced the provision of MC (Table 6).

## 4. Discussion 

According to this research, the level of counseling services provided by pharmacies in Pakistan is not up to par similar to pharmacies of India, Nigeria, Qatar, and Saudi Arabia, [7,32,33,34]. The rate of counseling without request was 3.6% which increased to 48% on demand. According to the empirical analyses, the rate of counseling was found to exhibit significant heterogeneity across the sample, ranging between 8% to 100%. More than half of SCs in both scenarios were advised to visit a physician. The present rate of referral surpasses that of prior studies performed in Malaysia, Slovenia, and Germany [3,35,36].

Considering the age of patients in both scenarios, ADI can be life threatening and need very careful use of antibiotics [16,17,37] but only 1.4% drug sellers can predict the potential ADIs as few of them advised the patients for medication change. The drug vendors in Pakistan lack knowledge about ADIs, similar to retail pharmacists in Virginia [38]. This indicates that drug vendors have inadequate training in providing guidance on the usage of antibiotics [23]. A pharmacist who has received sufficient training and knowledge keeps a close eye on antibacterial treatments and provides guidance for their proper administration [39]. Similarly, few SCs (1.4%) received special warnings or precautions about antibiotic use. However, these data are better than findings of a previous study from Qatar (0.0%) but much lower than a study from Germany (18.0%), Saudi Arabia (10.0%), and Sweden (20.5%) [3,4,25,40]. it was noted in this study that there was a scarcity of conversations regarding the potential adverse outcomes of therapy as described recently in a study of an Asian country [41].

A majority of the pharmacies did not ensure confidentiality whilst providing MC services. According to reports from SCs, the duration of the conversation ranged from 1 to 6 min, and in majority of cases, the allotted time was not enough, similar pattern is observed in a study from Saudi Arabia [42], similar result are also reflected in previous research of South Korean pharmacies [1]. In certain drugstores, employees allocate over 5 min towards MC services as found in previous studies of Malaysian pharmacies. Effective counseling can typically be accomplished in less than five minutes [3,43].

Non compliance to medication regimen and withdrawal of antibiotics leads to AMR. Short duration therapy of antibiotics is less effective and causes more resistance [14]. As found in a previous Malaysian study [43], this study indicates that pharmacy staff has no focus to demonstrate the importance of compliance. The importance of medication compliance was not emphasized and was not even mentioned in most cases. Non-compliance to antibacterial therapy and wrong storage of antibiotics is a leading cause of AMR rise in Pakistan [44]. Moreover, no information about drug storage was provided by the pharmacists in this study. The liquid antibiotics for children need refrigeration to maintain efficacy [45,46]. Most customers are also interested to know exact storage for antibiotics [28]. This investigation also demonstrates that pharmacy staff engaged in counseling has a good understanding of dietary, lifestyle modification and dose. In contrast to Ethiopian pharmacist, drug seller of Pakistan are least focused on the route of drug administration [46]. These evidences indicate the lack of many important aspects of counseling that ensure optimal use of antibiotics. The counseling provision is influenced by the number of waiting customers and age of pharmacy retailer like an Ethiopian study [47]. 

Effective communication plays a crucial part in advancing patient-centered practice and MC services within community pharmacies [28]. Effective communication is lacking in Pakistan. The provision of written material enhances patient satisfaction [48], but the present findings show that this is the weakest area in the counseling services of Pakistani pharmacies. Insufficient counseling practices and weak communication skills may be a result of lack of trainings and qualified supervision [22].

## 5. Conclusions

Community pharmacies in Pakistan focus only on antibiotic dispensing and trading. The staff’s scientific knowledge is very poor and their counseling skills require improvement. Antibiotic storage, route of administration, antibiotic–drug interactions, and side effects are not discussed. The effects of other medications and precautions about an antibacterial therapy are usually over-looked. Medical authorities should implement strategies to develop community pharmacy services based on modern scientific practice. In addition, the continuous training of staff would help to strengthen counseling.

## 6. Limitations

The present study has some limitations. First, this study was conducted in selected pharmacies. Although we tried to select a range of pharmacies from different areas, different results may have been found by selecting other pharmacies in other areas. Second, although rater reliability was analyzed, use of the SC method may have produced bias or misperceptions in assessing retailer age, provision of privacy, conversation time, and communication skills. Hence, the experience of retailers can’t be identified by this method. To overcome any potential bias, a score range was used to determine retailer age and conversation time and retailer communication skills were measured using a 5-point Likert scale. Similarly, a 3-point scale was used to determine privacy. Third, medicines were not available in some pharmacies, which may have influenced the results. Unavailability of medicine is a universal phenomenon. This factor can be eliminated by any method in any setting. Therefore, overall findings may vary on availability of medicines in all pharmacies. Fourth, the assessment of conversation time did not reflect the exact counseling time; the SCs were advised to estimate how long they had talked with staff about medication and disease. Finally, all the scenarios were performed by male SCs, as males are more dominant in Pakistani society. Therefore, this may have affected the results.

## Figures and Tables

**Figure 1 antibiotics-12-00931-f001:**
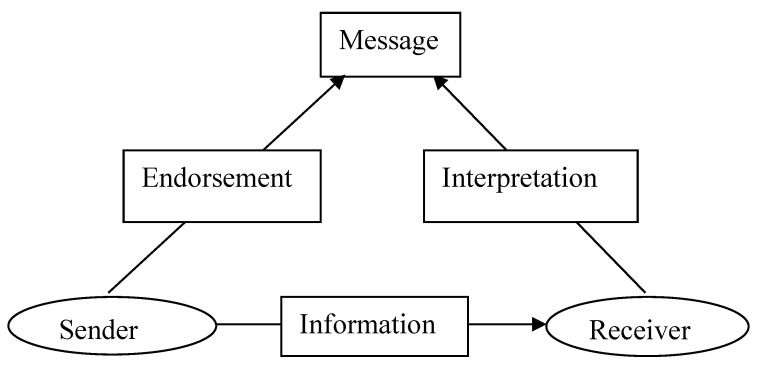
Ritual model of counseling.

**Table 1 antibiotics-12-00931-t001:** Specification of pharmacies, staff and visits.

Characteristics	Category	Scenario 1 (*n* = 554)	Scenario 2 (*n* = 465)	Overall %
		*n* (%)	*n* (%)	
Class of pharmacy (Number of the sale counters reported by SCs)	1	334 (60.3)	287 (61.7)	61
	2–5	204 (36.8)	164 (35.3)	36.1
	>5	16 (2.9)	14 (3.0)	2.9
Number of employees at the time of visit as reported by SCs	1–5	389 (70.2)	335 (72.1)	71.2
	6–10	156 (28.1)	125 (26.8)	27.5
	>10	9 (1.6)	5 (1.1)	1.4
Gender of pharmacy retailers (contacted)	Male	554 (100.0)	465 (100.0)	100.0
	Female	0 (0.0)	0 (0.0)	0.0
Age of contacted retailer (years), estimated by SC	<25	19 (3.4)	21 (4.5)	3.9
	25–35	164 (29.6)	153 (32.9)	31.3
	36–45	112 (20.2)	140 (30.1)	25.2
	46–55	202 (36.5)	123 (26.5)	31.5
	>55	57 (10.3)	28 (6.0)	8.2
Day of the visit	Monday	38 (6.8)	65 (13.9)	10.4
	Tuesday	59 (10.6)	42 (9.0)	9.8
	Wednesday	110 (19.8)	86 (18.5)	19.1
	Thursday	79 (14.2)	32 (6.9)	10.6
	Friday	58 (10.5)	101 (21.7)	16.1
	Saturday	66 (11.9)	70 (15.2)	13.6
	Sunday	144 (25.9)	69 (14.8)	20.4
Time of visit	8:00–12:00	146 (26.4)	167 (35.9)	31.1
	12:00–14:00	35 (6.3)	67(14.4)	10.4
	14:00–22:00	373 (67.3)	231 (49.7)	58.5
Number of waiting customers	0–5	361 (65.2)	338 (72.7)	68.9
	6–10	184 (33.2)	120 (25.8)	29.5
	>10	9 (1.6)	7 (1.5)	1.6
Waiting time (mints)	1–5	224 (40.4)	232 (49.9)	45.2
	>5	330 (59.6)	233 (50.1)	54.8
Total time of conversation (talk about medication) (mints)	≤2	311 (56.1)	284 (61.0)	58.6
	3–5	234 (42.2)	175 (37.6)	39.9
	>5	9 (1.7)	6 (1.3)	1.5
Privacy during conversation	Full provided	1 (0.2)	0 (0.0)	0.09
	Semi provided	17 (3.1)	4 (0.9)	2.0
	Not provided	536 (96.7)	461 (99.1)	97.9

**Table 2 antibiotics-12-00931-t002:** Response of selected pharmacies to request of medication.

Response		Scenario 1 (*n* = 554) *n* (%)	Scenario 2 (*n* = 465) *n* (%)	Overall %
Counseling without demand		2 (0.3)	32 (6.8)	3.6
On demand counseling	Provided	256 (46.2)	204 (43.8)	45
	Questioning and advised to contact a doctor	179 (32.3)	184 (39.5)	35.9
	Directly advised to contact a doctor	117 (21.1)	45 (9.6)	15.4

**Table 3 antibiotics-12-00931-t003:** Information acquired by staff from SCs.

Questions Asked	Scenario 1 (*n* = 437) *n* (%)	Scenario 2 (*n* = 420) *n* (%)	Overall %
Duration of disease or therapy	301 (68.8)	97 (23.0)	45.9
Whether had taken medicine before	195 (44.6)	161 (38.3)	45.5
Why the medicine was prescribed	-	91 (21.6)	21.6 ^*1^
Any co-morbidity or its treatment?	287 (65.6)	81 (19.2)	42.4
Any allergy to medicine in history?	79 (18.0)	55 (13.0)	15.5
Do you know how to take medicine?	15 (3.4)	76 (18.0)	10.7
You want to ask anything else?	3 (0.7)	18 (4.2)	2.5

*1. Scenarios 1 don’t have focus on this information; to avoid the biasness overall values don’t include its values.

**Table 4 antibiotics-12-00931-t004:** Contents of medication counseling provided to SCs.

Information Provided	Scenario 1 (*n* = 258) *n* (%)	Scenario 2 (*n* = 236) *n* (%)	Overall%
The name of the medicine discussed	-	13 (5.5)	5.5 ^*1^
Drug storage	0 (0.0)	0 (0.0)	0.0
How to take the medication (e.g., with/before/after meal)	7 (2.7)	12 (5.1)	3.9
Discussed root of drug administration	0 (0.0)	9 (3.8)	1.9
Dose of therapy	219 (84.9)	185 (78.4)	81.6
Duration of therapy	107 (41.4)	173 (73.3)	57.4
Possible side effects of therapy	2 (0.8)	3 (1.3)	1.1
Possible drug-drug interaction	4 (1.5)	3 (1.3)	1.4
Life style modification or dietary instruction	113 (43.8)	153 (64.8)	54.3
Any other special warnings or precautions about medication	4 (1.5)	3 (1.3)	1.4
Non compliance to medication and its effect	0 (0.0)	0 (0.0)	0.0
Effect of medicine withdrawal	0(0.0)	0(0.0)	0.0
Other medication during this therapy	0 (0.0)	0 (0.0)	0.0
Reinsured that client understand all instructions	49 (18.9)	16 (6.8)	12.8
Advice to change medication	4 (1.5)	3 (1.2)	1.4

*1. The name of medicine is mentioned by SC;therefore we escaped its values from final results.

**Table 5 antibiotics-12-00931-t005:** Communication skills of counseling person (Mean + SD).

Skill Used	Scenario 1 (*n* = 258)	Scenario 2 (*n* = 236)	Overall (*n* = 494)
Eye contact	2.3 ± 0.4	2.2 ± 0.5	2.25 ± 0.4
Attention to customer	2.5 ± 0.4	2.3 ± 0.4	2.4 ± 0.4
Engagement of customer	2.2 ± 0.3	2.2 ± 0.3	2.2 ± 0.3
Non-verbal expressions used	1.4 ± 0.2	1.4 ± 0.3	1.4 ± 0.2
Provided written information	1.1 ± 0.3	1.1 ± 0.4	1.1 ± 0.3

**Table 6 antibiotics-12-00931-t006:** Factors influencing the provision of medication counseling (*p* values).

Factors	Scenario 1	Scenario 2
Number of employees at the time of visit	0.294	0.387
Age of pharmacy retailer (contacted)	0.013	0.656
Day of visit	0.178	0.614
Time of visit	0.627	0.952
Number of waiting customers	0.834	0.043

*p* values less than 0.05 was consider statically significant.

## Data Availability

No new data were created or analyzed in this study. Data sharing is not applicable to this article.

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
