# Peer review of "Evaluating the Counseling Standards and Ability of Pharmacy Staff to Detect Antibiotic-Drugs Interactions: A Simulated Client Study from Pakistan"

_antibiotics, 2023, doi:10.3390/antibiotics12050931_

Round 1
Reviewer 1 Report
Dear Author,
The article is poorly written and you can submit it to any local journal. I have found many serious flaws in the article.
Author Response
The article is poorly written and you can submit it to any local journal. I have found many serious flaws in the article.
Thank you very much for your professional comments to enhance the quality of the manuscript. It's re-written, I hope now it meets the criteria for journals.
Reviewer 2 Report
The authors should consider the followings:
1. The authors should clearly explain and evaluate the inter-rater reliability.
2. Was the amount of SCs enough for the study? On average, how many pharmacies did each SC assessed?
3. Please supplement the pilot study data as supplementary information for this article.
4. "A list of pharmacies was obtained from Department of Health" were the list up-to-date? When were the last census of pharmacies for the list generation?
5. Please provide information on when did this study perform (the duration), and the factor(s) affected by Covid-19, if during peak seasons of covid-19 and if any.
6. Please clarify the novelty of the current study in the abstract and conclusion part of the article.
7. Please kindly explain why the Ethical approval was needed regarding the institute, “Health Science Center of Xi’an Jiaotong University (Ref # MR102-15/Phar)”, while the study is based on Pakistan population?
8. Were there inter-rater differences, between fresher Pharm D students, or graduates with multi-year of experiences?
9. "All SCs were trained well" Please list the criteria of technical competencies of the SCs.
10. Did the assessment involve audio recording? If yes, did the audio recording aid to grade the inter-rater reliability?
11. The author should show the results regarding the inter-rater reliability.
12. Was there a plan for the assessment time (hour of the day, i.e. peak hour of that pharmacy)? And would the time of assessment of that pharmacy affecting the overall result?
13. Did the results reflect any concern of quality regarding the peak hour vs normal hour of the working condition?
14. The authors can elaborate more on what they define, response distribution (60%).
15. Would the study also consider, the customer flow (services amount) of each pharmacy, compared to their serving population?
16. In sample size calculation, did the author consider the potential drop-out rate?
17. The authors should explain the rationale(s) for choosing 4% as margin of error, in the sample size calculation.
18. In Table 1, Day of the visit, did the study mark any public holiday? and the interference of the public holiday to the study?
19. Please give rationale(s) of why the contacted gender of pharmacy retailers (all male), also that the SCs were all male, as well.
20. In Table 6, please provide detailed rationales, regarding the items on statistical significance in Table 6, i.e. Age of pharmacy retailer (contacted), Number of waiting customers.
Author Response
Thank you very much for your professional comments to enhance the quality of the manuscript. The questions are addressed and incorporated in the manuscripts as detailed below.
The authors should clearly explain and evaluate the inter-rater reliability
Explained in Subsection of the pilot study.
Three experts including a senior community pharmacist, a pharmacy academician and sociologist assessed the scenario presentation of SC. Their performance was rated and scored.
Was the amount of SCs enough for the study? On average, how many pharmacies did each SC assessed?
Each SC visited 27 pharmacies but ONLY "3 pharmacies/ day" were visited
Please supplement the pilot study data as supplementary information for this article.
Supplementary file is attached.
"A list of pharmacies was obtained from Department of Health" were the list up-to-date? When were the last census of pharmacies for the list generation?
In Punjab of Pakistan, every pharmacy is registered through Department of Health to start its business. So record is automatically updated on day by day basis
Please provide information on when did this study perform (the duration), and the factor(s) affected by Covid-19, if during peak seasons of covid-19 and if any.
This cross-sectional study was performed between June 2016 and February 2017. Study was conducted before the covid-19 , So no pandemic effect influences this study.
Please clarify the novelty of the current study in the abstract and conclusion part of the article
Incorporated in the Abstract section.
Please kindly explain why the Ethical approval was needed regarding the institute, “Health Science Center of Xi’an Jiaotong University (Ref # MR102-15/Phar)”, while the study is based on Pakistan population?
As this project was a part of my PhD studies from Health Science Center of Xi’an Jiaotong University. So it's ethical permission was necessary. To overcome further confusion for readers. this portion is eliminated from revised manuscript.
Were there inter-rater differences, between fresher Pharm D students, or graduates with multi-year of experiences?
As this is simulated patient's study, we can't differentiated. This is added in the limitation portion
"All SCs were trained well" Please list the criteria of technical competencies of the SCs
Three experts including a senior community pharmacist, a pharmacy academician and sociologist assessed the scenario presentation of SC. Their performance was rated and scored.
Did the assessment involve audio recording? If yes, did the audio recording aid to grade the inter-rater reliability?
According to the infrastructure of community pharmacies in Pakistan, audio recording can't give clear results. Moreover, ethical considerations didn't allow the Audio or video recording. So we didn't apply any health technology assessment.
The author should show the results regarding the inter-rater reliability
Given in the supplementary results
Was there a plan for the assessment time (hour of the day, i.e. peak hour of that pharmacy)? And would the time of assessment of that pharmacy affecting the overall result?
Peak hours of pharmacies varies with respect to locations and season. However, most were visited at routine time ( day time)
Did the results reflect any concern of quality regarding the peak hour vs normal hour of the working condition?
The quality of counseling not varies with respect to time. Peak times are not defined . No significant difference was found with respect to time, as assessed.
The authors can elaborate more on what they define, response distribution (60%)
The response distribution 60% was applied due to probability of distribution variable and possible variability in responses (added in manuscript )
Would the study also consider, the customer flow (services amount) of each pharmacy, compared to their serving population?
Pharmacy distribution in Pakistan is not according to population.
In sample size calculation, did the author consider the potential drop-out rate?
In SC method probability of drop is absolutely zero. So we didn't consider it.
We consider the drug availability as given in limitation portion of study.
The authors should explain the rationale(s) for choosing 4% as the margin of error, in the sample size calculation.
To determine the maximum accuracy in the sample margin of error was 4% (added in the manuscript )
In Table 1, Day of the visit, did the study mark any public holiday? and the interference of the public holiday to the study?
The only holidays in the community pharmacy business (in Pakistan) are festivals i.e Eid ul Fiter and Eid Ul Azha. So no pharmacy was visited during this public holidays.
Please give rationale(s) of why the contacted gender of pharmacy retailers (all male), also that the SCs were all male, as well.
It's a male dominant conservative society, the due to addition of female, results may be effected. To avoid the gender bias , only male SC were selected. As there is the concern of male retailers, the pharmacies in Pakistan have minimal representation of female staff and ownership.
In Table 6, please provide detailed rationales, regarding the items on statistical significance in Table 6, i.e. Age of pharmacy retailer (contacted), Number of waiting customers.
Elaborated in the discussion section
Regards
Reviewer 3 Report
This is a well written article, about an important issue.
The section on theories should be moved to the introduction.
Table 1 is not essential for the paper, it could be described in the body of the results section.
Author Response
The section on theories should be moved to the introduction.
Thank you very much for your professional comments to enhance the quality of the manuscript. Moved to the introduction section.
Regards
Round 2
Reviewer 1 Report
Dear Authors
I have found serious flaws with work, i fear it may mislead the scientific community.